# Differential Effects of D9 Tetrahydrocannabinol (THC)- and Cannabidiol (CBD)-Based Cannabinoid Treatments on Macrophage Immune Function In Vitro and on Gastrointestinal Inflammation in a Murine Model

**DOI:** 10.3390/biomedicines10081793

**Published:** 2022-07-26

**Authors:** Zhanna Yekhtin, Iman Khuja, David Meiri, Reuven Or, Osnat Almogi-Hazan

**Affiliations:** 1Laboratory of Immunotherapy and Bone Marrow Transplantation, Hadassah Medical Center, The Faculty of Medicine, Hebrew University of Jerusalem, Jerusalem 91120, Israel; zhannay@hadassah.org.il (Z.Y.); iman.khuja@mail.huji.ac.il (I.K.); reuvenor@hadassah.org.il (R.O.); 2The Laboratory of Cancer Biology and Cannabinoid Research, Department of Biology, Technion—Israel Institute of Technology, Haifa 320003, Israel; dmeiri@technion.ac.il

**Keywords:** cannabinoid, cannabis, immune, macrophage, elderly, inflammatory bowel disease, cannabidiol, D9 tetrahydrocannabinol, nitric oxide

## Abstract

Phytocannabinoids possess a wide range of immune regulatory properties, mediated by the endocannabinoid system. Monocyte/macrophage innate immune cells express endocannabinoid receptors. Dysregulation of macrophage function is involved in the pathogenesis of different inflammatory diseases, including inflammatory bowel disease. In our research, we aimed to evaluate the effects of the phytocannabinoids D9 tetrahydrocannabinol (THC) and cannabidiol (CBD) on macrophage activation. Macrophages from young and aged C57BL/6 mice were activated in vitro in the presence of pure cannabinoids or cannabis extracts. The phenotype of the cells, nitric oxide (NO•) secretion, and cytokine secretion were examined. In addition, these treatments were administered to murine colitis model. The clinical statuses of mice, levels of colon infiltrating macrophages, and inflammatory cytokines in the blood, were evaluated. We demonstrated inhibition of macrophage NO• and cytokine secretion and significant effects on expression of cell surface molecules. In the murine model, clinical scores were improved and macrophage colon infiltration reduced following treatment. We identified higher activity of cannabis extracts as compared with pure cannabinoids. Each treatment had a unique effect on cytokine composition. Overall, our results establish that the effects of cannabinoid treatments differ. A better understanding of the reciprocal relationship between cannabinoids and immunity is essential to design targeted treatment strategies.

## 1. Introduction

Macrophages are specialized innate immune cells that orchestrate homeostatic, inflammatory, and reparative activities. Murine macrophages are located in the brain, skin, liver, kidney, lungs, and heart and originate from the yolk sac or fetal liver; their maintenance in adulthood in the absence of stressors is independent of circulating monocytic precursors. In other tissues, such as the gastrointestinal tract, monocytic precursors contribute to tissue macrophages [1]. In the steady state, tissue macrophages have intrinsic anti-inflammatory functions. Tissue stress, including infection, drives the production of monocytes and neutrophils. Bone marrow-derived monocytes are recruited to the damaged site, differentiate into macrophages and dendritic cells, and begin the inflammatory processes [2]. These events must be tightly regulated. Dysregulation of macrophage differentiation and function is involved in the pathogenesis of different diseases, including inflammatory bowel disease (IBD).

Immunological dysregulation in IBD is characterized by epithelial damage, expansion of inflammation driven by intestinal flora, a large number of cells infiltrating into the lamina propria, and a failure of immune regulation to control the inflammatory response [3]. In IBD patients, the number of macrophages increase in the inflamed mucosa [3].

The endocannabinoid system (ECS) regulates various aspects of physiological, behavioral, immunological, and metabolic functions. It is now clear that many of the components of the endocannabinoid system function as key regulators of the immune system and the immune response [4]. Endocannabinoid ligands and receptors are involved in the regulation of both innate and adaptive immune cells. Murine and human monocytes/macrophages and microglial cells express the endocannabinoid receptors CB1 and CB2. CB2 receptors in macrophages have anti-inflammatory properties [5,6,7,8], while CB1 receptors have proinflammatory properties and are involved in phagocytosis [9,10,11,12]. Importantly, the expression levels of cannabinoid receptors in leukocytes are influenced by different inflammatory factors [13].

Phytocannabinoids, the biologically active constituents of cannabis, possess a wide range of immune regulatory properties, mediated by the endocannabinoid system. Two cannabinoids have been the focus of most of the studies that have examined medical uses, i.e., D9 tetrahydrocannabinol (THC) and cannabidiol (CBD). THC and some of the other phytocannabinoids mediate their biological effects primarily through the classical cannabinoid receptors CB1 and CB2. In addition, THC can act as an agonist of the receptors/channels GPR55, GPR18, PPARγ, transient TRPA1, TRPV2, TRPV3, and TRPV4, and as an antagonist of the receptors/channels TRPM8 and 5-HT3A. Interestingly, although CBD affects the immune function, it has a very weak affinity to CB2 or CB1, where it can act as a negative allosteric modulator. Several reports have demonstrated that CBD acts as an agonist of other receptors/channels, such as TRPA1, TRPV1, TRPV2, TRPV3, PPARγ, and 5-HT1A, and as an antagonist of the receptors GPR55, GPR18, and 5-HT3A. CBD is also an inverse agonist of the receptors GPR3, GPR6, and GPR12 [4].

Previously, we compared the influence of cannabinoid-based treatments on lymphocyte function [14]. The aim of the current research was to examine the consequences of treatment with THC and CBD on macrophage activation and in macrophage-related inflammation. Since THC and CBD mediate their actions on mammal cells though different receptors, we hypothesized that each cannabinoid has selective effects on macrophage phenotype and function, and hence, a different impact on activation and inflammation. Therefore, the aim of our research was to elucidate the differential effects of THC- and CBD-based treatments on macrophage immune function. Our previous results suggested that the combination of cannabinoids with other active molecules in the plant may achieve better clinical results than pure cannabinoids, therefore, we also examined the differences between the effects of high THC and high CBD cannabis extracts (Table 1).

While Cannabis is not yet registered as a drug, the potential of cannabinoid-based medicines for the treatment of various conditions has led many countries to authorize their clinical use. As a result, in recent years, there has been a rapid increase in the medical use of cannabis and a wide range of cannabinoid-based treatments are offered to patients. THC and CBD are considered to be the two essential elements in these treatments. Therefore, it is crucial to explore the various biological effects of these molecules. A better understanding of the effects of THC, CBD, and other active molecules on the immune response will assist physicians in providing the best possible individually targeted treatment for their patients and will allow the design of new treatments.

## 2. Materials and Methods

### 2.1. Cannabis Extracts and Cannabinoids

This research was performed under the approval of The Medical Cannabis Unit in the Israeli Ministry of Health (REQ46). Pure THC was generously provided by the laboratory of Prof. Raphael Mechoulam. Synthetic CBD was purchased from STI Pharmaceuticals Ltd., Newtown, UK. Cannabis Sativa and Indica extract with high content in THC or CBD (i.e., THCE/CBDE, respectively) were supplied by Cannabliss (Cannabliss Ltd., Tel Aviv, Israel). Extraction was obtained using ethanol, and evaporated. Identification and quantification of phytocannabinoids in the cannabis extracts were done by ultrahigh performance liquid chromatography with an ultraviolet detector (UHPLC/UV) system (Thermo Scientific, Bremen, Germany). The terpenoid analysis was performed by static headspace gas chromatography tandem mass spectrometry (SHS-GC/MS/MS) using full evaporation technique with external calibrations, as previously described [15,16]. The main molecules are listed in Table 1.

### 2.2. Mice

Female and male 8- to 11-week-old and 8-month-old C57BL/6 female mice were purchased from Envigo, Jerusalem, Israel and were acclimated for at least 7 days before the experiment in the specific pathogen-free (SPF) facility of the Authority of Biological and Biomedical Models at the Hebrew University of Jerusalem. The aged mice were up to 18 months old in the SPF animal facility. The study was approved by the Institutional Animal Care and Use Committee of the Hebrew University of Jerusalem in accordance with national laws and regulations for the protection of animals (MD-22-16868-4, MD-20-16432-4, and MD-18-15565-5). The mice were housed under specific SPF conditions in the animal facility under the AAALAC accreditation, throughout the experiments.

### 2.3. Peritoneal Macrophages

Peritoneal exudate cells were induced in mice by an intraperitoneal injection of 0.5 mL of 3% thioglycollate (BD DIFCO, Franklin Lakes, NJ, USA). After 4 days, mice were anesthetized with ketamine and xylazine, and then killed by cervical dislocation. Peritoneal exudate cells were washed from the peritoneal cavity of mice by lavage with 5 mL of ice-cold, sterile phosphate buffered saline (PBS). Cells were washed with PBS and re-suspended in Dulbecco’s modified Eagle medium (DMEM) (Sartorius, Israel) supplemented with 10% fetal calf serum (FCS), 1% penicillin/streptomycin, and 1% L-glutamine (Biological industries/Sartorius, Beit Haemek, Israel). Cell viability was determined by MTT colorimetric assay in which a yellow tetrazole, is reduced to purple formazan in living cells (MP Biomedicals, LLC, Solon, OH, USA). The resultant color was measured at 450 nm using a Biotek PowerWave XS Microplate Reader.

### 2.4. Nitric Oxide (NO•) Determination

Peritoneal macrophages were seeded at a density of 2.5 × 10^5^ cells/well in 96-well plates and incubated overnight at 37 °C and 5% CO_2_. On the following day, the medium was changed to fresh DMEM containing 5 µg/mL CBD, THC, or cannabis extracts. The cells were then stimulated for 24 h by the addition of lipopolysaccharide (LPS) to a concentration of 1 μg/mL. After 24 h, cell supernatants (SNs) were harvested for nitric oxide radical (NO•) assay by addition of 100 μL SN to an equal volume of Griess reagent (1% sulfanilamide, 0.1% naphthalene diamine, and 2% H_3_PO_4_). After 10 min of incubation, the resultant color was measured at 550 nm. The amount of NO• produced, and any inhibition by the tested materials, was calculated from a standard curve prepared with NaNO_2_. Controls: non-activated cells, activated cells + vehicle, activated cells + 1400W dihydrochloride (NOS2 inhibitor, Enzo Life Sciences Inc., Lausen, Switzerland).

### 2.5. Flow Cytometry

5 × 10^5^ cells/sample were washed once in ice-cold staining buffer (PBS containing 1% FBS, pH 7.2). Then, cells were stained in the dark at 4 °C for 30 min with fluorochrome-labeled anti-mouse mAb (Biolegend, San Diego, CA, USA), specific for cell surface antigens: F4/80, I-Ad (MHC class II), and CD16/32. Cells were subsequently washed, re-suspended in staining buffer, and analyzed by flow cytometry.

### 2.6. RNA Extraction and Real-Time PCR Analysis

Total cellular RNA was extracted using RNeasy Mini Kit columns (Geneaid, New Taipei City, Taiwan), according to the manufacturer’s protocol. One microgram of total RNA was used to synthesize cDNA using a high-capacity cDNA kit (Applied Biosystems, Waltham, MA, USA), following the supplier’s instructions. Detection of transcript levels of CB1 and CB2 was performed using a TaqMan Gene Expression Assay Kit (Applied Biosystems, Waltham, MA, USA), with HPRT-1 as a reference. All primers were purchased from Applied Biosystems (Waltham, MA, USA). Real-Time PCR reactions were conducted using a QuantStudio 5 instrument (Applied Biosystems, Waltham, MA, USA). Data were analyzed using the QuantStudio design and analysis Software (Applied Biosystems, Waltham, MA, USA).

### 2.7. Induction of Colitis in Mice

Colitis was induced in C57BL/6 mice with 2% DSS dissolved in drinking water given ad libitum (Days 1–7), and then replaced with plain drinking water for 3 days. Then, 5 mg/kg cannabis/cannabinoids were prepared in 5% Cremophor EL (Sigma, St. Louis, MO, USA), 5% ethanol (Gadot, Haifa, Israel) in PBS, and 0.1 mL were administered IP every other day, starting from Day 1. Body weight and stool were monitored once a day. Changes of body weight are indicated as loss of baseline body weight (% of initial weight). Clinical score (0–9) included: stool score (0–3), rectal score (0–3), and general clinical parameters (fur texture, behavior, and posture, 0–3). On the tenth day of colitis induction, blood was collected from the mouse tails into ethylenediaminetetraacetic acid (EDTA)-coated capillary tubes, and then the mice were anesthetized using ketamine and xylazine, and then killed by cervical dislocation. The intestines were excised, measured, and carefully rinsed with saline. Blood tubes were centrifuged at 1500 rpm, room temperature, for 5 min; plasma was collected and kept at −80 °C for cytokine and chemokine analysis.

### 2.8. Histopathology and Immunohistochemistry

Colon tissue was fixed in 4% buffered formaldehyde (Bio-Lab, Jerusalem, Israel) and embedded in paraffin. For histology, the sections were stained with H&E according to standard protocols. Histological scoring (0–9) was based on 3 parameters: Crypt damage (0–3), percent involvement (0–3), and damage to bowel wall structure (0–3).

For immunostaining, paraffin embedded sections were heated to 60 °C, deparaffinized using xylene, dehydrated using ethanol, and washed with H_2_O. Sections were treated with 3% H_2_O_2_ and antigens retrieved by incubation with 1 mg/mL pronase. Then, the samples were washed in PBS and blocked in CAS blocking reagent (Rhenium, Modi’in, Israel). The slides were stained with anti-F480 (Bio-Rad, Hercules, CA, USA). Anti-rat IgG universal immune peroxidase polymer (Nichirei Biosciences Inc., Tokyo, Japan) was used as secondary antibody. Sections were incubated with Stable Peroxidase Substrate Buffer (Thermo Scientific, Waltham, MA, USA), washed with H_2_O, and analyzed on a BX41 microscope (Olympus Corporation, Tokyo, Japan).

### 2.9. Proinflammatory Chemokine and Cytokine Analysis

Peritoneal macrophages were activated for 24 h with LPS, in the presence of 5 µg/mL CBD, THC, or cannabis extracts. The supernatant was collected and analyzed using a LEGENDplex™ MU Macrophage/Microglia Panel cytokine array assay (Biolegend, San Diego, CA, USA), according to the manufacturer’s instructions.

Plasma samples from DSS model mice were analyzed using a LEGENDplex™ Mouse Inflammation Panel cytokine array (Biolegend, San Diego, CA, USA), according to the manufacturer’s instructions.

### 2.10. Statistical Analysis

Data from in vitro studies are represented as mean ± SE. The mean was calculated from the indicated number of experiments. The mean of triplicates from each experiment was used for this calculation. For statistical analysis of the macrophage NO• secretion experiments in female and male mice, we used Friedman test. For statistical analysis of the macrophage NO• secretion and cannabinoid receptors expression, macrophage cytokine secretion, and flow cytometry experiments in young and aged mice, we used the Mann–Whitney test. For statistical analysis of colon length, colon histopathology, macrophage infiltration to the colon, and blood cytokines in the DSS model experiments, we used the Kruskal–Wallis test. For statistical analysis of weight loss and clinical score in the DSS model experiments, we used the Kruskal–Wallis test of area under the curve (AUC). In all experiments, *p* value < 0.05 were considered statistically significant.

## 3. Results

### 3.1. Cannabinoid Treatments Reduce Nitric Oxide and Cytokine Production of LPS-Activated Peritoneal Macrophages

Upon activation, macrophages produce large amounts of nitric oxide (NO•). To test the effect of cannabinoid treatments on macrophage activation, NO• secretion was determined. Macrophages from previously thioglycollate (tg)-injected C57BL/6 mice were collected by peritoneal lavage, and then activated for 24 h with lipopolysaccharide (LPS), in the presence of cannabinoid treatments; 1400W dihydrochloride, a specific iNOS inhibitor, served as control. All treatments show dose dependent effect on NO• secretion (Appendix A). For our further experiments, we used 5 µg/mL of each treatment (THC, CBD, or cannabis extracts). Our results demonstrate 42–72% inhibition of activation-induced NO• secretion from peritoneal macrophages from female (Figure 1a) and male (Figure 1b) mice in the presence of cannabinoid treatments. The reduced NO• secretion was not caused by decreasing cell number, since the treatments showed no toxic effect on the cells in MTT viability assay (Appendix A). The differences between THC and CBD treatments were significant only in the female mice. Importantly, both extracts were significantly more efficient (*p* < 0.0001) than the pure cannabinoids in female and male mice. A treatment with the combination of THC and CBD (2.5 µg/mL of each) was less effective than the pure cannabinoids (Appendix A).

Next, we examined the influence of aging on the responsiveness of macrophages to cannabinoid treatments. For this aim, we obtained peritoneal macrophages from aged (18 months old) mice and compared their NO• secretion with macrophages from young (2 months old) mice. Figure 1c (left) shows elevated secretion of NO• from non-activated cells and reduced secretion upon activation of old peritoneal macrophages. The effect of cannabinoid treatments on NO• secretion was significantly reduced as compared with cells from young mice (Figure 1c, right). In addition, peritoneal macrophages from aged mice demonstrate alleviated expression of the cannabinoid receptors (Figure 1d).

To examine the effect of the treatments on inflammatory cytokine/chemokine secretion from peritoneal macrophages, we collected the supernatant of 24 h LPS-activated cells and analyzed the levels of different cytokines using a LEGENDplex™ MU Macrophage/Microglia Panel cytokine array assay. IL6, TNF-alpha, CXCL2, and G-CSF levels in the culture media increased upon activation (Figure 2a–d). Our results demonstrate 40–74% inhibition of activation-induced IL6 secretion from peritoneal macrophages (Figure 2a), 22–66% inhibition of TNF-alpha secretion (Figure 2b), 4–44% inhibition of CXCL2 secretion (Figure 2c), and 0–58% inhibition of GM-CSF secretion (Figure 2d). CBD and CBDE had more significant inhibitory effect on cytokine/chemokine secretion as compared with THC and THCE. The differences between CBD and THC are significant in TNF-alpha, CXCL2, and G-CSF. IL12p40, CCL22, and IL18 were also elevated following activation, however, the cannabinoid-based treatments did not have clear effects on their levels (Appendix A).

### 3.2. Cannabinoid Treatments Affect the Phenotype of Activated Peritoneal Macrophages

To learn more about CBD and THC molecular effects in macrophages, we tested cell surface expression of several molecules in the activated macrophages. Class II molecules of the major histocompatibility complex (MHCII) is upregulated on some polarized macrophage populations upon activation. We found that both THC and CBD treatments induce small, but significant, further elevation in MHCII expression (Figure 3a). R. A. Ezekowitz and S. Gordon have demonstrated that expression levels of the F4/80 glycoprotein and the Fc receptor CD16/32 on peritoneal macrophages are dependent on the activator [17]. In our study, in tg-induced macrophages activated with LPS, F4/80 was elevated by THC (15%), but reduced by CBD treatment (32%) (Figure 3b). CD16/32 cell surface expression was elevated on LPS-activated cells, but the cannabinoid treatments had no significant effect on its expression (Figure 3c).

### 3.3. Cannabis Extracts Have Improved Effect in Murine Colitis DSS Model Mice as Compared with Pure Cannabinoids

We chose the murine colitis dextran sodium sulfate (DSS) model to compare the efficacy of the different cannabinoid treatments in macrophage-related inflammation in vivo. DSS is a chemical colitogen with anticoagulant properties. In the acute intestinal inflammation model, DSS causes disruption of the intestinal epithelial monolayer lining, leading to the entry of luminal bacteria and associated antigens into the mucosa and activation of the innate immunity [18].

Acute colitis was induced by adding DSS to the mice drinking water at 2% (*w*/*v*) ad libitum for 7 days, and then replaced with plain water. Cannabinoid treatments, 5 mg/kg, were administered intraperitoneal, every other day, from Day 1 to Day 10 (Figure 4a). All treatments, but particularly the CBD extract (CBDE), significantly inhibited weight loss in the DSS mice (Figure 4b). THC and CBD extracts both had better effect on the disease clinical score as compared with the pure cannabinoids (Figure 4c). This improved effect was also evident in the measurement of colon length (Figure 4d); the average length of the colon in the CBDE-treated mice was 4.8 cm, significantly higher than the average length of the colon in the CBD-treated mice 3.9 cm (*p* < 0.0001). In THCE-treated mice, the average length of the colon was 4.8 cm vs. 4.1 cm in the THC-treated mice (*p* < 0.0001). All treatment significantly improved the clinical condition of the mice as compared with the vehicle DSS group. The difference between the CBD and THC groups was not significant.

The improved clinical outcome of all cannabinoid-based treatments is also demonstrated in the histopathology of the colon (Figure 5a,b). Colon sections were stained with haematoxylin/eosin and scored for the infiltration with inflammatory cells, damage in crypt architecture, and thickening of the bowel wall. In this assay, no significant differences between the four cannabinoid-based treatments were found.

### 3.4. Cannabinoid Treatments Reduce Intestinal Macrophage Infiltration and the Levels of Inflammatory Cytokines in the Plasma of DSS Mice

Next, we examined the effects of the treatments on macrophages and inflammation in the DSS model. In IBD patients, an increased number of macrophages in the inflamed mucosa initiate a rapid response to luminal microbial antigens [3]. Immunohistochemistry with F4/80 antibody was performed to detect macrophages in the colon. As demonstrated in Figure 6a,b, the number of macrophages in the colon tissue is highly elevated in the DSS mice. All the cannabinoid-based treatments significantly inhibited colon infiltration of macrophages in the DSS mice (40–60% inhibition). CBDE treatment inhibition of macrophage infiltration was significantly more effective than pure CBD treatment (*p* = 0.03). The THCE and THC treatments were not significantly different.

Cytokine levels in the blood reflect the inflammatory status in the body. It was also demonstrated that inflammatory cytokines have a crucial role in the pathogenesis of IBD, where they control multiple aspects of the inflammatory response [19]. We, therefore, tested the plasma levels of inflammatory cytokines of treated DSS mice using a mouse inflammation cytokine array assay. Interestingly, we found that the different treatments have unique effects on plasma cytokines (Figure 7). IL6 levels were reduced by all treatments, but particularly by CBDE. The levels of TNF-alpha were significantly reduced only by the cannabis extracts; but pure cannabinoids had stronger effect on IFN-beta levels. The levels of other tested cytokines were not significantly changed in the plasma of DSS mice as compared with healthy controls.

## 4. Discussion

Macrophages play key roles in innate immunity. They are specialized cells involved in the detection and destruction of bacteria and other harmful pathogens. In addition, they can present antigens to T cells and initiate inflammation. Macrophages and microglial cells express the Gi protein-coupled seven transmembrane cannabinoid receptors, CB1 and CB2 [12,20,21,22,23,24]. On the one hand, CB2 receptors in macrophages have anti-inflammatory properties, can affect macrophage polarization, and are involved in promoting autophagy [3,5,6,8]. CB1 receptors in macrophages, on the other hand, have proinflammatory properties and are involved in phagocytosis [9,10,11,12].

Phytocannabinoids possess a wide range of immune regulatory properties, mediated by the endocannabinoid system [4]. Two cannabinoids have been the focus of most of the studies examining medical uses: THC and CBD. These two phyto-cannabinoids utilize different endocannabinoid receptors to mediate their effects. A few studies have demonstrated anti-inflammatory effect for phytocannabinoid treatments on macrophage function, however, these studies have generally focused on a single cannabinoid [25,26,27,28,29].

Previously, we compared the influence of THC, CBD, and cannabis extracts on lymphocyte activation in vitro and in a murine graft versus host disease (GvHD) model [14]. In the current research, we aimed to compare the consequences of treatment with THC or CBD on macrophage activation and in DSS murine model for gastrointestinal inflammation. We use this model as a model for acute inflammation with involvement of macrophages in the pathophysiology of the disease [19].

First, we examined the effect of cannabinoid treatments on NO• secretion from peritoneal macrophages, ex vivo. Macrophages produce large amounts of NO• as a defense mechanism. However, in pathological conditions, such as autoimmune diseases, there is excessive simultaneous production of NO• and O_2_•− by macrophages and other cells. NO• and O_2_•− generate large quantities of the toxic molecule peroxynitrite (ONOO−), an oxidant and nitrating agent which can damage a wide array of molecules in cells, including DNA and proteins [30]. Elevated NO• bioavailability has been implicated in the etiology of a number of pathological events including IBD. It is now clear that the ECS plays a key role in regulating NO• formation via CB1R, CB2R, and/or alternative molecular targets [31]. We found that both THC and CBD, as well as cannabis extracts treatments, inhibit activation-induced NO• secretion from both female and male peritoneal macrophages. These results were in agreement with the results of Romano et al. who demonstrated a similar effect with the cannabinoid THCV [32]. Cannabis extracts and pure cannabinoids were used in a concentration of 5 µg/mL; the extracts had stronger effect than the pure cannabinoids, although cannabinoids constituted only 35–38% of their content. This could result either from inhibitory signaling of other molecules in the plant (not THC/CBD) or from a synergistic function of THC/CBD with other molecules. Interestingly, CBD signaling has been associated with inhibition of cytokine/chemokine secretion, since both pure CBD and CBDE had greater effect on the levels of peritoneal macrophage’s secreted cytokines as compared with THC and THCE. It should be noted that the cannabis extracts had no significant advantage over the pure cannabinoids in inhibition of cytokine/chemokine secretion from peritoneal macrophages. The inhibitory effect of CBD, on the one hand, IL6 and TNF-alpha, and on the other hand, CXCL2 and G-CSF may suggest its ability to regulate both local inflammation and periphery bone marrow interactions.

Aging is concurrent with a slow and constant functional deterioration of the immune system, known as immunosenescence, which is accompanied by an increase in chronic inflammatory processes, a phenomenon known as “inflammaging”. The devastating consequences of an aged immune system include impairment of the ability to cope with infections and an increased risk of developing chronic diseases such as cancer and heart disease—the two leading causes of death in old age [33]. Indeed, our results demonstrate elevated NO• secretion from non-activated peritoneal macrophages from aged mice, which indicated pre-existing inflammatory process. In addition, the reduced secretion upon LPS activation of aged mice peritoneal macrophages, may indicate immunosenescence. The old cells also showed decreased responsiveness to cannabinoid treatments and alleviated expression of CB1 and CB2 receptors. These results corresponds with earlier studies which demonstrated age-related alterations in the endocannabinoid system [34,35], and may be accompanied with other age-related changes in the endocannabinoid receptors and enzymes in the cells.

The effect of THC and CBD treatments on the expression of cell surface molecules on activated peritoneal macrophages was examined. Stimulation with LPS, a major component of the outer membrane of Gram-negative bacteria, leads to an effective classical, proinflammatory, macrophage activation. MHC-II molecules, expressed by antigen presenting cells, are central in the initiation of cellular and humoral immune responses. Upon LPS activation, MHCII is upregulated in macrophages [36], and hence increased antigen presentation. MHCII levels can also be affected by different soluble factors, such as cytokines [37]. In a previous study by Paul W. Wacnik et al. THC reduced MHCII expression in LPS-activated dendritic cells [38]. Since cannabinoid treatments reduced nitric oxid secretion, we expected similar reduced expression of MHCII. Surprisingly, we found that both treatments did not reduce, but induced, a small elevation of MHCII expression in the activated macrophages. The combination of these results may indicate that treatment of cannabinoids together with LPS may lead to M2b polarization of the peritoneal macrophages. M2b macrophages are a population of regulatory macrophages, classically induced by LPS and immune complexes, G-protein coupled receptor ligands, or prostaglandins [39,40].

F4/80 is widely used as a mouse macrophage antigen marker. Although the ligands for F4/80 have not been defined, evidence shows that this molecule is required for peripheral tolerance [41]. In addition, it has been demonstrated that F4/80 and Fc receptor CD16/32 expression levels in peritoneal macrophages are dependent on the activator [17]. In our experiments, the expression of F4/80 was elevated by THC, but reduced by CBD treatment as compared with the activated control cells. This result demonstrates the different consequences of the treatment with different cannabinoids. However, the functional relevance of this result is yet to be determined. CD16 and CD32 are low affinity IgG Fc receptors. Immune complexes signaling through the Fc receptors can lead to M2B polarization [39]. In our experiments, CD16/32 cell surface expression was elevated in LPS-activated cells, however, the cannabinoid treatments had no significant effect on their expression. Therefore, it is possible that the cannabinoids bypass CD16/32 signaling, for example, by binding to PPARγ, which may provide the second signal for M2B polarization.

IBD is comprised of well-known gastrointestinal autoimmune inflammatory disorders such as ULcerative colitis (UC) and Crohn’s (CD). Cannabinoid receptor stimulation attenuates murine colitis, while cannabinoid receptor antagonism and cannabinoid receptor deficient models reverse these anti-inflammatory effects [4]. Phytocannabinoids have been used in preclinical models of gastrointestinal inflammation [42,43] and several clinical trial have tested the efficacy of cannabinoid-based treatments in IBD patients [42,44]. Indeed, in our DSS murine colitis model, cannabinoid treatments improved the clinical condition of the mice. Similar to the in vitro nitric oxide secretion assay and to our previous results in the murine GvHD model [14], the results of treatment with cannabis extracts were superior to pure-cannabinoid treatments. This unique effect could be a result of either a synergistic function of THC/CBD with other components or from independent anti-inflammatory properties of other molecules in the plant.

Dysregulated macrophages have a key role in the pathogenesis of IBD. In IBD patients, the number of macrophages increases in the inflamed mucosa. The phenotype and functions of the macrophages in the inflamed sites differ from those in physical conditions. For instance, they express high levels of costimulatory molecules and produce high levels of IL-12 and IL-23 in vitro under the microbial stimulation [3]. Our results demonstrate that cannabinoid-based treatments inhibit colon infiltration of macrophages in DSS mice. The average number of colon infiltrating macrophages in each treatment group correlated with the severity of the disease.

Inflammatory cytokines are produced by macrophages, but also by other cells such as dendritic cells, fibroblasts, and lymphocytes. The cannabinoid treatments all affected the levels of inflammatory cytokines in the mice blood. However, each treatment had a unique effect on the composition of cytokines: CBD treatment reduced the levels of IL6 and the more regulatory cytokine, interferon-beta; THC treatment reduced the levels of IL6 and interferon-beta better than CBD; CBDE treatment reduced the levels of TNF-alpha and had the strongest effect on the level of IL6; and THCE treatment reduced the levels of IL6, interferon-beta; and TNF-alpha. These results demonstrate, once again, the different consequences of treatment with different cannabinoid-based treatments. Interestingly, IL6, a key cytokine in IBD [19] was reduced by all the cannabinoid treatments, but TNF-alpha, another cytokine involved in the pathogenesis of IBD, was significantly affected, in the murine model, only by the cannabis extracts. Therefore it is possible that the reduced levels of TNF-alpha are responsible for the improved clinical effect. Importantly, anti-TNF agents are often used for IBD treatment in the clinic, due to the central role of this cytokine in the pathogenesis of the disease in human subjects too. CBDE was the most effective treatment in IL6 and TNF-alpha inhibition, both in activated peritoneal macrophages and in the murine model. This result was in agreement with the recently published results of Aswad et al., who demonstrated similar effects for a different high-CBD cannabis extract in human PBMCs and neutrophils and in murine models for systemic and lung inflammation [45].

## 5. Conclusions

Overall, our results indicate both similarities and differences between the impact of CBD- and THC-based drugs. Although all the tested treatments had an anti-inflammatory effect, their specific effects (for example, on phenotype of the cells and on cytokine production) differed. These differences may influence the clinical outcome of the treatment. We were surprised to find very similar anti-inflammatory results for the two cannabis extracts, which had diverse content of THC and CBD. This could suggest that THC/CBD content may not be the best indicator for anti-inflammatory properties of a cannabis-based drug. These results highlight the need to expand the research on the interplay between cannabinoids and other phytochemicals in the cannabis extracts. A better understanding of the effects of each molecule and the synergism between these molecules on the immune response will assist physicians to provide the best possible individually targeted treatment for their patients and will allow the design of new treatments.

## Figures and Tables

**Figure 1 biomedicines-10-01793-f001:**
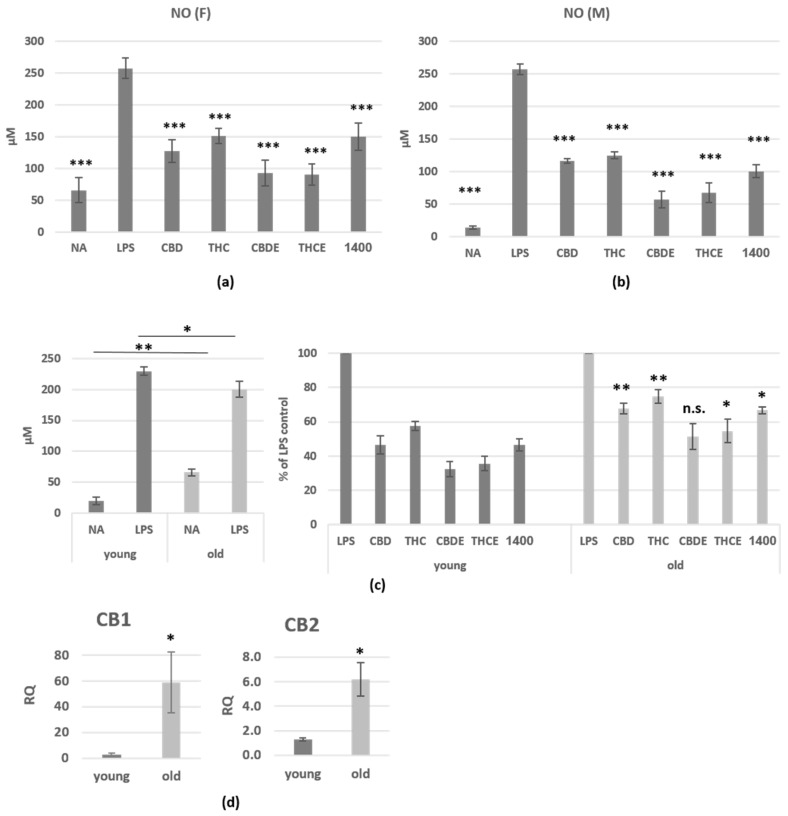
The influence of pure CBD/THC and cannabis extracts on nitric oxide production of LPS-activated peritoneal macrophages. Peritoneal macrophages from C57BL/6 female (**a**) and male (**b**) mice were activated for 24 h with LPS, in the presence of cannabinoid treatments (5 µg/mL). 1400W (1400), a specific iNOS inhibitor, served as control. NO• levels in the supernatant were analyzed. (**a**)—*n* = 10 mice, from 3 independent experiments; (**b**)—*n* = 7 mice, from 3 independent experiments. The differences of all treatments as compare with LPS-activated control (indicated on the graphs) are highly significant. The differences of THCE and THC between CBDE and CBD are significant in (**a**,**b**); (**c**) NO• levels in the supernatant of activated peritoneal macrophages from young (2 months old) and aged (18 months old) C57BL/6 female and male mice. *n* = 7 mice per group, from 3 independent experiments. The differences between aged and young mice in each treatment, are indicated on the graph; (**d**) the expression levels of CB1 and CB2 in peritoneal macrophages from young and aged C57BL/6 female mice (*n* = 3/group) were assessed by real-time PCR analysis. The results are expressed as mean + SEM. *p*-value *, <0.05; **, <0.01; ***, <0.001. NA—non-activated; LPS—lipopolysaccharide-activated macrophages; THC—D9 tetrahydrocannabino; CBD—cannabidiol; THCE—high THC cannabis extract; CBDE—high CBD cannabis extract; n.s.—not significant.

**Figure 2 biomedicines-10-01793-f002:**
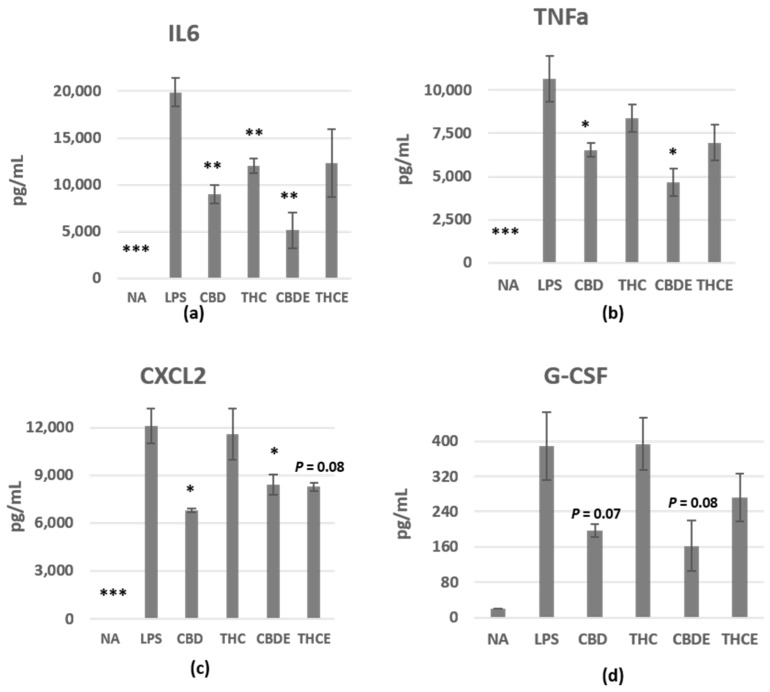
The influence of pure CBD/THC and cannabis extracts on cytokine/chemokine production of LPS-activated peritoneal macrophages. Peritoneal macrophages were activated for 24 h with LPS, in the presence of cannabinoid treatments (5 µg/mL), as in Figure 1. *n* = 3 mice. IL6 (**a**), TNF-alpha (**b**), CXCL2 (**c**), and G-CSF (**d**) levels in the culture supernatant were detected. The results are expressed as mean + SEM. *p*-value as compare with LPS-activated control cells *, <0.05; **, <0.01; ***, <0.001. NA—non-activated; LPS—lipopolysaccharide-activated macrophages; THC—D9 tetrahydrocannabinol; CBD—cannabidiol; THCE—high THC cannabis extract; CBDE—high CBD cannabis extract.

**Figure 3 biomedicines-10-01793-f003:**
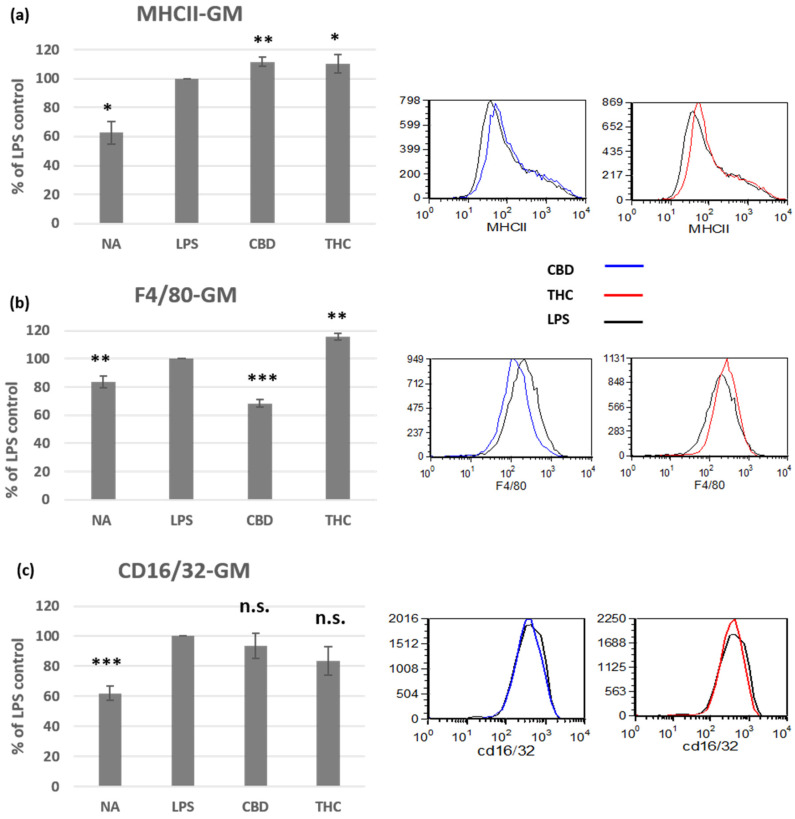
The effect of CBD and THC on the phenotype of LPS-activated peritoneal macrophages. Peritoneal macrophages from C57BL/6 female mice were activated for 24 h with LPS, in the presence of cannabinoid treatments (5 µg/mL). Cell surface expression levels of MHCII ((**a**), *n* = 5), F4/80 ((**b**), *n* = 6), and CD16/32 ((**c**), *n* = 7) were determined by flow cytometry. The results are expressed as mean + SEM. *p*-value as compare with LPS-activated control cells *, <0.05; **, <0.01; ***, <0.001. NA—non-activated; LPS—lipopolysaccharide-activated macrophages; THC—D9 tetrahydrocannabinol; CBD—cannabidiol; n.s.—not significant.

**Figure 4 biomedicines-10-01793-f004:**
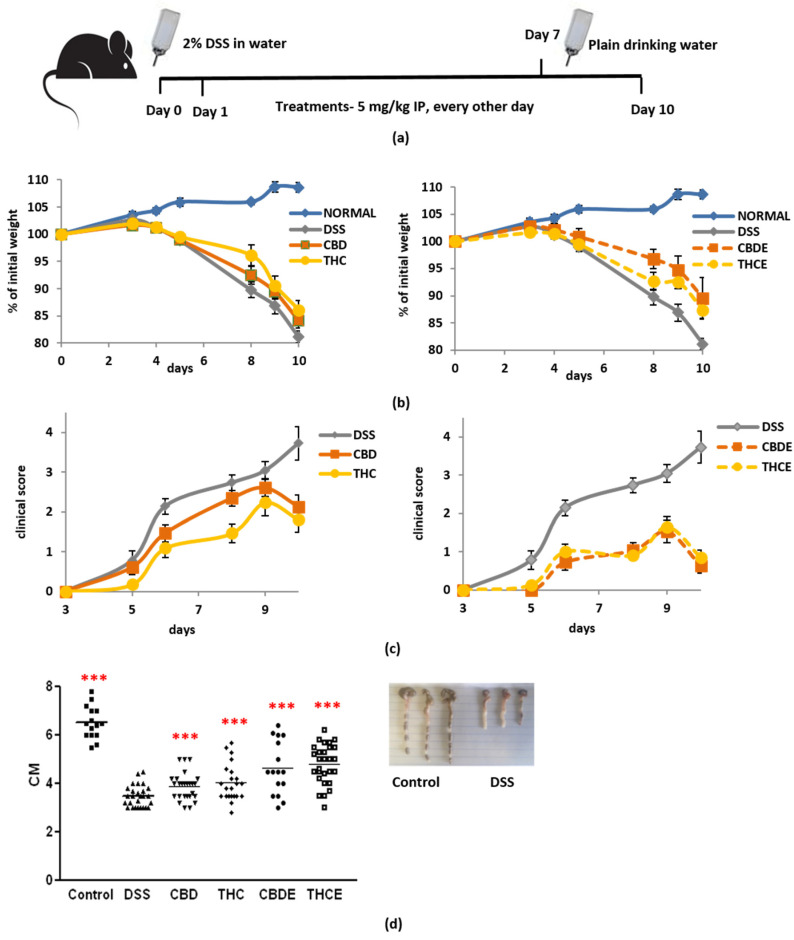
Cannabis/cannabinoids administration for treatment of murine colitis in DSS model mice: (**a**) Acute colitis was induced by adding DSS to the mice drinking water at 2% (*w*/*v*) ad libitum for 7 days, and then replaced with plain water. 5 mg/kg cannabis/cannabinoids were administered IP every other day, starting from Day 1; (**b**) the effect of pure cannabinoids (left) and cannabis extracts (right) on weight loss. Differences between DSS group and THCE- as well as CBDE-treated groups are significant. The difference between the CBD- and CBDE-treated groups is significant *p* < 0.05; (**c**) the effect of pure cannabinoids (left) and cannabis extracts (right) on clinical score. Differences between the DSS group and all cannabinoid-treated groups are significant *p* < 0.005. Differences between cannabis extracts and pure cannabinoids are significant *p* < 0.008. The difference between the CBD- and THC-treated groups is significant *p* < 0.05; (**d**) the effect of pure cannabinoids and cannabis extracts on colon length, at the end of the experiment (Day 10). *p*-value as compare with the DSS control group ***, <0.001. The differences between THCE and THC and between CBDE and CBD are significant. Data are summarized from 5 independent experiments, 5–7 mice/group in each experiment. DSS—dextran sodium sulfate; THC—D9 tetrahydrocannabinol; CBD—cannabidiol; THCE—high THC cannabis extract; CBDE—high CBD cannabis extract.

**Figure 5 biomedicines-10-01793-f005:**
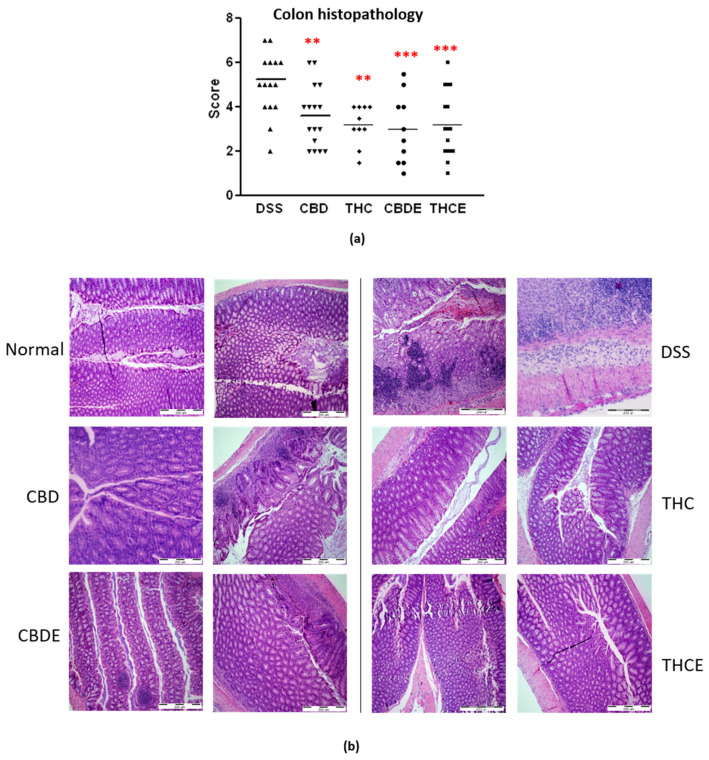
Histopathology of the colons of DSS model mice. Paraffin sections were stained with H&E and scored for inflammation and tissue damage; (**a**) Summary of average scores. *p*-value as compare with the DSS control group **, <0.01; ***, <0.001; (**b**) representative pictures of H&E stained sections. DSS—dextran sodium sulfate; THC—D9 tetrahydrocannabinol; CBD—cannabidiol; THCE—high THC cannabis extract; CBDE—high CBD cannabis extract.

**Figure 6 biomedicines-10-01793-f006:**
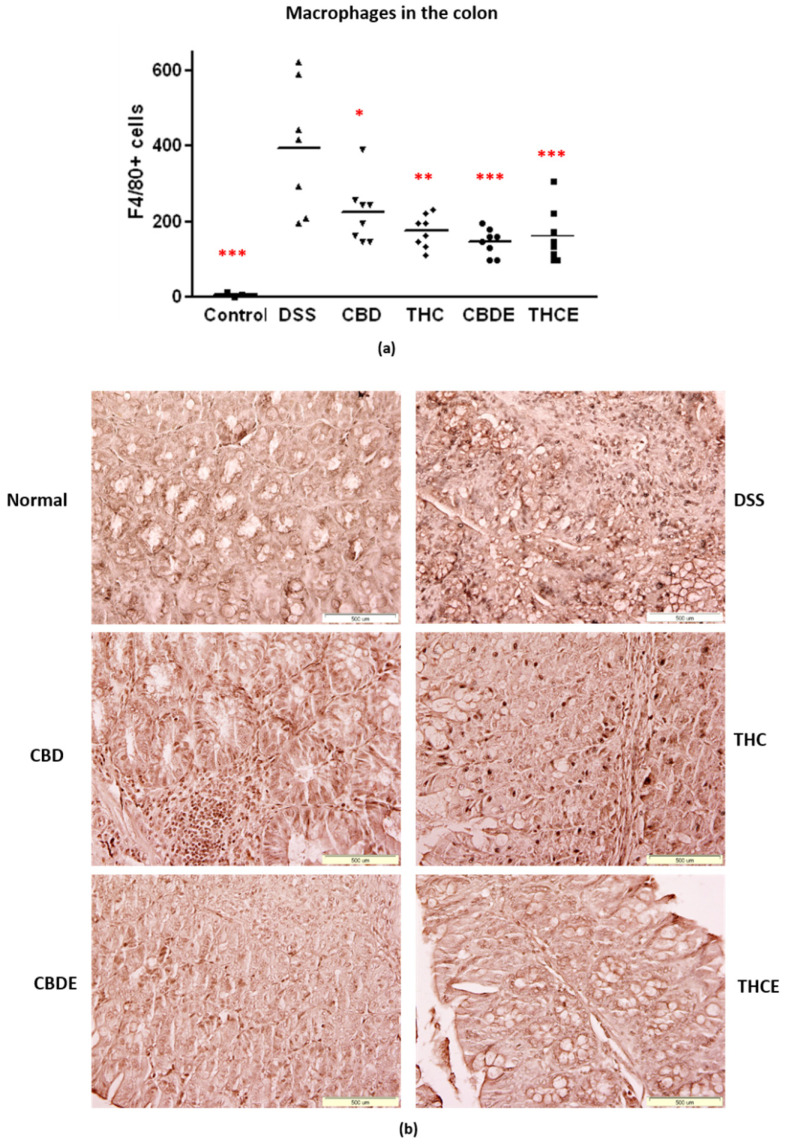
Immunostaining for macrophages in the colons of DSS model mice. Paraffin sections were stained with anti-F4/80 antibodies: (**a**) Average number of positive cells. *p*-value as compare with the DSS control group *, <0.05; **, <0.01; ***, <0.001. The difference between the CBDE and CBD groups is significant; (**b**) representative pictures of F4/80 immunostained sections. DSS—dextran sodium sulfate; THC—D9 tetrahydrocannabinol; CBD—cannabidiol; THCE—high THC cannabis extract; CBDE—high CBD cannabis extract.

**Figure 7 biomedicines-10-01793-f007:**
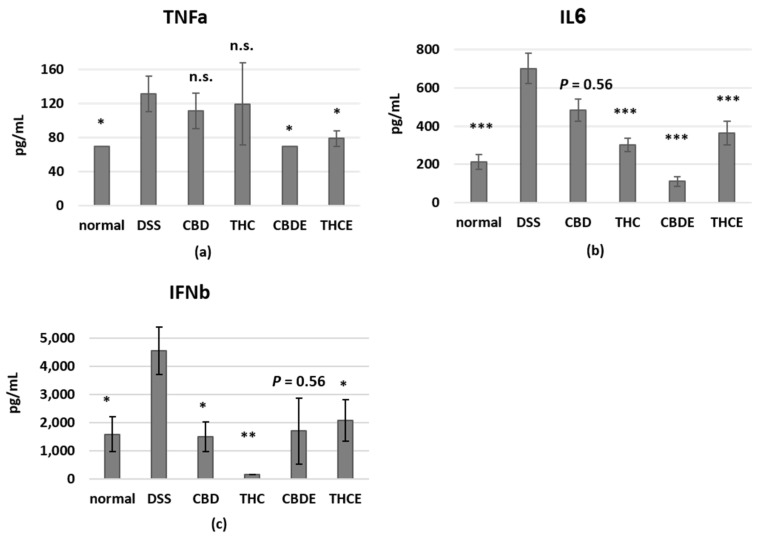
The influence of pure CBD/THC and cannabis extracts on inflammatory cytokines in the blood of DSS model mice. Blood samples for cytokine analysis were obtained at Day 10. The levels of TNFa (**a**), IL6 (**b**), and IFNb (**c**) in the plasma were determined using a LEGENDplex™ Mouse Inflammation Panel cytokine array; *n* = 5–10 mice/group. The results are expressed as mean + SEM. *p*-value as compare with the DSS group *, <0.05; **, <0.01; ***, <0.001. DSS—dextran sodium sulfate; THC—D9 tetrahydrocannabinol; CBD—cannabidiol; THCE—high THC cannabis extract; CBDE—high CBD cannabis extract; n.s.—not significant.

**Table 1 biomedicines-10-01793-t001:** Chemical analysis of the main phytocannabinoids and terpenoids in the cannabis extracts.

		THCE	CBDE
**Phytocannabinoids (%)**	Total THC	**24.58**	1.3488
HPLC-UV	Total CBD	9.62	**36.0906**
	Total CBG	0.3	0.4412
**Terpenoids (ppm)**	Linalool	346.1	1087.0
SHS-GC/MS/MS	Fenchyl alcohol	850.9	924.1
	α-Terpineol	825.8	992.0
	β-Caryophyllene	1548.7	695.3
	α-Humulene	406.9	265.2

## Data Availability

Not applicable.

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
