# Peer review of "Differential Effects of D9 Tetrahydrocannabinol (THC)- and Cannabidiol (CBD)-Based Cannabinoid Treatments on Macrophage Immune Function In Vitro and on Gastrointestinal Inflammation in a Murine Model"

_biomedicines, 2022, doi:10.3390/biomedicines10081793_

Round 1
Reviewer 1 Report
In this manuscript Yekthin et al investigate the effect of THC, CBD as well as of THC- ad CBD -enriched extracts in the activation of macrophages in vitro and in a model of gastrointestinal inflammation.
The main idea of this manuscript is exploring the differences exerted by pure THC and pure CBD vs extracts enriched in these two cannabinoids that also contain other cannabinoids/plant components and that therefore could potentially contribute to modulate/enhance the effects exerted primarily by THC and/or CBD
General major concerns:
1. From the methodological point of view the major concern with this study is that it does not indicate the actual doses/concentrations of THC and CBD that are present in the extracts as compared with the doses of pure THC and CBD used in the experiments. The authors indicate generically (not in the figure legends but in the discussion) that “Cannabis extracts and pure cannabinoids were used in concentration of 5μg/ml”. To establish these comparisons in a precise manner authors should perform dose-response experiments using different concentrations of THC, CBD and of extracts containing equivalent concentrations of these cannabinoids, so one can analyze whether a similar amount of THC or CBD present in the extract produces (or not) a different effect than the same dose of one (or the combination of the two) cannabinoids. Likewise, the authors do not explore the effect of using a mixture of pure THC and pure CBD to see whether this combination produces a different effect than the extracts that contain the same concentration of these two cannabinoids. Overall, the study is designed and represented in a way that precise comparisons between the effects triggered by pure cannabinoids and cannabinoid-enriched extracts are not possible. Since the effects of THC ad CBD alone on macrophages have been already described a much more thorough pharmacological analysis is in my opinion essential.
2. The other major concern with the study is perhaps more conceptual. Beyond the methodological problems described above, the article is mainly descriptive. The authors speculate that the differences found between the effect of the treatment with pure cannabinoids and cannabinoid-enriched extracts are due to the so-called “entourage effect”. However, the study does not explore what is the pharmacological explanation for this entourage effect. For example, as the authors discuss, it is assumed that the effects of THC are mainly due to activation of CB2 and CB1 receptors in macrophages whereas the effect of CBD could be based on the stimulation of other receptors. Based on these observations, could antagonist of these receptors prevent equally the effects of pure cannabinoids and THC or CBD-enriched extracts? What are the precise cannabinoids/terpenoids present in these extracts that modify the effect of THC and CBD? What are the pharmacological targets of these cannabinoids/terpenoids? The authors discuss that “a better understanding of the effects of each molecule on the immune response will assist physicians in providing the best possible individually targeted treatment for their patients and will allow the design of new therapies”. Unfortunately, at least as it stands, this study is mainly descriptive and does not shed light on this issue. In my opinion, for these observations to be truly useful for the physicians it will be essential to define pharmacologically which are the main active components of these extracts, which are their molecular targets and how they produce a different effect than THC or CBD.
Author Response
We thank the reviewer for his/her comments, which certainly improved our manuscript.
- Comment: The main idea of this manuscript is exploring the differences exerted by pure THC and pure CBD vs extracts enriched in these two cannabinoids that also contain other cannabinoids/plant components and that therefore could potentially contribute to modulate/enhance the effects exerted primarily by THC and/or CBD.
- Answer: While Cannabis is not yet registered as a drug, the potential of cannabinoid-based medicines for the treatment of various conditions has led many countries to authorize their clinical use. As a result, in recent years, there has been a rapid increase in the medical use of cannabis and a wide range of cannabinoid based treatments are offered to patients. Currently, in most of these treatments, THC and CBD are considered the two essential elements in these treatments. Our study was designed to compare the effects of THC and CBD. For this aim, on the one hand we used pure THC vs. pure CBD, and on the other, high THC extract versus high CBD extract. It appears that the description of the design was not clear. We have corrected this in the introduction. The experimental design was not intended to make any comparison between plant extract and pure cannabinoids, however we cannot overlook the actual results, which demonstrate better anti-inflammatory efficacy for both extracts over the pure cannabinoids in the models we have used.
- Comment: From the methodological point of view the major concern with this study is that it does not indicate the actual doses/concentrations of THC and CBD that are present in the extracts as compared with the doses of pure THC and CBD used in the experiments. The authors indicate generically (not in the figure legends but in the discussion) that “Cannabis extracts and pure cannabinoids were used in concentration of 5μg/ml”. To establish these comparisons in a precise manner authors should perform dose-response experiments using different concentrations of THC, CBD and of extracts containing equivalent concentrations of these cannabinoids, so one can analyze whether a similar amount of THC or CBD present in the extract produces (or not) a different effect than the same dose of one (or the combination of the two) cannabinoids. Likewise, the authors do not explore the effect of using a mixture of pure THC and pure CBD to see whether this combination produces a different effect than the extracts
- Answer: The concentration of the treatments was indicated both in the Methods and in the Results sections. As suggested by the reviewer, we have also added it to the figure legends. We have also added a graph demonstrating reduced activity of 2.5μg/ml CBD + 2.5μg/ml THC combination as compared to 5μg/ml of a single cannabinoid, in NO secretion assay (Fig S3). Our preliminary results demonstrated that all the treatments have dose dependent inhibitory effect on NO secretion. Following the reviewer's recommendation, we have added these results to the supplementary data (Fig S1). We chose to use 5μg/ml of each treatment since in this concentration THC and CBD are effective and not toxic to the cells. To use extract with similar cannabinoid content, we would have needed to increase their concentration X3 and disregard all the other active molecules.
- Comment: The study does not explore what is the pharmacological explanation for this entourage effect. For example, as the authors discuss, it is assumed that the effects of THC are mainly due to activation of CB2 and CB1 receptors in macrophages whereas the effect of CBD could be based on the stimulation of other receptors. Based on these observations, could antagonist of these receptors prevent equally the effects of pure cannabinoids and THC or CBD-enriched extracts? What are the precise cannabinoids/terpenoids present in these extracts that modify the effect of THC and CBD? What are the pharmacological targets of these cannabinoids/terpenoids? The authors discuss that “a better understanding of the effects of each molecule on the immune response will assist physicians in providing the best possible individually targeted treatment for their patients and will allow the design of new therapies”. Unfortunately, at least as it stands, this study is mainly descriptive and does not shed light on this issue. In my opinion, for these observations to be truly useful for the physicians it will be essential to define pharmacologically which are the main active components of these extracts, which are their molecular targets and how they produce a different effect than THC or CBD.
- Answer: Indeed, it is tempting to create a pharmacological formulation of cannabinoids and terpenes, which will have anti-inflammatory properties, comparable to the cannabis extracts. In our hands these formulations did not work as expected (in several in-vivo and in-vitro models). For this reason we are sharing the content of cannabinoids and terpenes in the extracts we have used in this research with the scientific community. Hopefully this information together with our pre-clinical observations will contribute to the development of such a drug. Meanwhile, since cannabis based treatments are already widely used (around 50,000 new patients/year in Europe according to the European Monitoring Centre for Drugs and Drug Addiction, EMCDDA, report, 2022), we believe that our data is of clinical importance. The interplay between phyto-cannabinoids and the endocannabinoid system in inflammation is a fascinating subject, which we are currently investigating.
Reviewer 2 Report
Cannabis reserach is a hot topic and this manuscript it is interesting. The covered area deserves to be investigated and published. In general, the manuscript is clear and reserach well conducted. However, there are some aspects that must be adressed before its publication, especially in the introduction. Here there are few comments that can help:
INTRODUCTION:
- In the introduction I suggest including the information of the receptors that commonly mediate CBD/THC effects. At least, their relationship with CB1 and CB2.
- Other points. CB2 receptors are known to be overexpressed in immunomodulatory cells. I suggest including this info.
- Lines 76-81. I think this paragraph are methods. It should not be included in the introduction. Lines 82-89. These are results!!! This can be in the abstract not in the introduction.
- I think that last paragraph does not provide any information.
- Instead of the paragraphs mentioned above, I suggest including a statement in which the clear objectives of the work are mentioned.
Methods:
- I guess that cannabis Indika refers to Cannabis Indica sp. Right? This reviewer always read it as Indica. Please, verify this.
- Line 107. Please verify 1,2. I think it is a typo and must be deleted.
- Section 2.3 Please, include more information. e.g volume of drugs, euthanasia method,….
- In section 2.9 Instead of mentioning the figures when explaining statistical test, it would be better to mention the type of experiment.
- I am curious, in which solvent did the authors administer cannabinoids? Due to its low aqueous solubility, it is challenging
Results and discussion:
- I am wondering, Fig.4 has included error bars?
- Did the authors evaluate the combination of pure CBD and pure THC? In my opinion, it could be a reason why the extracts are more active. Of course, it could be related to other minor compounds but in other therapeutical uses the combination of CBD and THC have demonstrated an entourage effect. I think they have different mechanisms of action that could be potentiate with their combination.
Author Response
We thank the reviewer for his/her comments, which certainly improved our manuscript.
INTRODUCTION:
- In the introduction I suggest including the information of the receptors that commonly mediate CBD/THC effects. At least, their relationship with CB1 and CB2.
We added a paragraph on the receptors that commonly mediate CBD/THC effects (line 68-77).
- Other points. CB2 receptors are known to be overexpressed in immunomodulatory cells. I suggest including this info.
We added a sentence on CB2 expression in leukocytes (lines 62-64).
- Lines 76-81. I think this paragraph are methods. It should not be included in the introduction. Lines 82-89. These are results!!! This can be in the abstract not in the introduction.
These parts were removed from the introduction.
- I think that last paragraph does not provide any information.
- Instead of the paragraphs mentioned above, I suggest including a statement in which the clear objectives of the work are mentioned.
These were replaced as suggested (lines 89-97).
Methods:
- I guess that cannabis Indika refers to Cannabis Indica sp. Right? This reviewer always read it as Indica. Please, verify this. The name was corrected in the text.
- Line 107. Please verify 1,2. I think it is a typo and must be deleted. This part was removed.
- Section 2.3 Please, include more information. e.g volume of drugs, euthanasia method,…. volume of drugs, euthanasia method were added to Sections 2.3 and 2.6
- In section 2.9 Instead of mentioning the figures when explaining statistical test, it would be better to mention the type of experiment. Section 2.9 was corrected.
- I am curious, in which solvent did the authors administer cannabinoids? Due to its low aqueous solubility, it is challenging. This was added to Section 2.6
Results and discussion:
- I am wondering, Fig.4 has included error bars? Error bars were added in Fig 4
- Did the authors evaluate the combination of pure CBD and pure THC? In my opinion, it could be a reason why the extracts are more active. Of course, it could be related to other minor compounds but in other therapeutical uses the combination of CBD and THC have demonstrated an entourage effect. I think they have different mechanisms of action that could be potentiate with their combination. We did this experiment. The results were added to the supplement as Figure S3.
Reviewer 3 Report
Zhanna and colleagues describes Differential effects of D9 tetrahydrocannabinol (THC) and cannabidiol (CBD) based cannabinoid therapies on macrophage immune function in-vitro and on gastrointestinal inflammation in a murine model. The authors then concluded that all treatments inhibit activation induced NO• secretion from macrophages, cannabis extracts have a stronger effect than the pure cannabinoids. The research presented in this manuscript is well rationalized, executed and interpreted. Importantly, their findings are highly consistent. Following are the suggestions to the authors that may improve the impact of this manuscript.
- Line 104 on page 3, Please elaborate the extraction process, as there is no information available. If extraction techniques are already published, then include in line 109, before evaporation. Add ‘full extraction’.
- In Section 2.2, please mention how to quarantine the mice after purchased from Envigo.
3. Section 2.3, Please add few lines about how Author’s sacrifice the animals?
4. Section 2.4, What is NaNO2? Please write the full form and accordingly H3PO4 as well and also, check thoroughly everywhere in the manuscript.
- Section 2.5 line 142, give space after pH7.2, check thoroughly in the manuscript and do it accordingly.
6. Section 2.6, ‘Blood tubes were centrifuged’, please mention at what speed the blood samples were centrifuged and what is the temperature, which instrument and specification of the instrument. Complete information is required here.
7. Section 2.7, How the Author’s perfused the colon tissue, please write the complete details.
8. Section 2.9 on line 188, give space after ‘mean± SE. Similarly in line 191, ‘Figures 1c,d’ give space after 1c. check thoroughly in the manuscript and do it accordingly.
9. Section 3.1, Author’s mentioned ‘male (Figure 1b), please include the study design in section 2.2 as there it is not describing about male mice. Please check once.
10. What is MTT viability assay, please describe in Materials and methods section.
11. Conclusion is missing, please add few lines about what are the significant findings and future perspectives.
Author Response
We thank the reviewer for his/her comments, which certainly improved our manuscript.
- Line 104 on page 3, Please elaborate the extraction process, as there is no information available. If extraction techniques are already published, then include in line 109, before evaporation. Add ‘full extraction’. The cannabis extracts were supplied by a cannabis company (Cannabliss LTD., as indicated in the text) and therefore we don’t have more detailed protocol on the cultivation and extraction processes.
- In Section 2.2, please mention how to quarantine the mice after purchased from Envigo. We added a section on mice quarantine in Section 2.2
- Section 2.3, Please add few lines about how Author’s sacrifice the animals? We added a mice sacrificing description.
- Section 2.4, What is NaNO2? Please write the full form and accordingly H3PO4 as well and also, check thoroughly everywhere in the manuscript. We have corrected this in the manuscript
- Section 2.5 line 142, give space after pH7.2, check thoroughly in the manuscript and do it accordingly.
- Section 2.6, ‘Blood tubes were centrifuged’, please mention at what speed the blood samples were centrifuged and what is the temperature, which instrument and specification of the instrument. Complete information is required here. We added the missing information.
- Section 2.8, How the Author’s perfused the colon tissue, please write the complete details. We added the missing information.
- Section 2.9 on line 188, give space after ‘mean± SE. Similarly in line 191, ‘Figures 1c,d’ give space after 1c. check thoroughly in the manuscript and do it accordingly.
- Section 3.1, Author’s mentioned ‘male (Figure 1b), please include the study design in section 2.2 as there it is not describing about male mice. Please check once. We added this description.
- What is MTT viability assay, please describe in Materials and methods section. The MTT assay is described in lines 206-209.
- Conclusion is missing, please add few lines about what are the significant findings and future perspectives. We have changed this part and it is included is lines 608-619 of the discussion.
Reviewer 4 Report
Cannabinoids are naturally occurring chemicals produced by the Cannabis sativa plant and biochemically classified as terpenophenols. Among the most important active components are: tetrahydrocannabinol (THC), primary active component cannabidiol (CBD) and cannabinol (CBN). As is known, some cannabinoids are also used in medical therapy with different indications. In this work, the effect of cannabinoid treatments on macrophage activation was tested, using macrophages from young and elderly C57BL/6 mice activated in vitro in the presence of pure cannabinoids or cannabis extracts.
The proposed topic is valid and current, in fact, the efficacy of some cannabinoids in medical therapy is currently recognized, even if their use in pharmacology is still debated.
The experimental part is well structured and supported by statistical analysis.
The references are sufficiently up to date.
However, the manuscript requires some modifications, which are outlined below.
Lines 82-96: Move these paragraphs to the results section.
Line125: Insert the full name and the acronym in brackets DMEM (Dulbecco's Modified Eagle Medium). Similarly, fetal calf serum (FCS).
I suggest that the authors include a list of major abbreviations at the end of the manuscript.
Furthermore, although the efficacy of cannabinoids as anti-asthmatics, or as antiemetics, in open-angle glaucoma, multiple sclerosis, anxiety, insomnia and depression is recognized, their use in pharmacology is still debated. Therefore, in the conclusions section, authors should refer to this aspect.
Author Response
We thank the reviewer for his/her comments, which certainly improved our manuscript.
Lines 82-96: Move these paragraphs to the results section. This part was removed from the introduction.
Line125: Insert the full name and the acronym in brackets DMEM (Dulbecco's Modified Eagle Medium). Similarly, fetal calf serum (FCS). Added
I suggest that the authors include a list of major abbreviations at the end of the manuscript. We added this at the end of the manuscript.
Furthermore, although the efficacy of cannabinoids as anti-asthmatics, or as antiemetics, in open-angle glaucoma, multiple sclerosis, anxiety, insomnia and depression is recognized, their use in pharmacology is still debated. Therefore, in the conclusions section, authors should refer to this aspect. We added this to the last paragraph of the introduction and to the discussion.
Round 2
Reviewer 1 Report
I acknowledge the authors’ effort in trying to answer to the different points raised in my report. I also understand that it would be necessary to perform additional experiments and therefore more time.to answer some of these points. Nonetheless, the main concerns raised in my original report remain to be solved.
Reviewer 3 Report
Accept in present form